# Patient propagules: Do soil archives preserve the legacy of fungal and prokaryotic communities?

**Gian Maria Niccolò Benucci** [1,2]*, **Bryan Rennick** [1], **Gregory Bonito** [1,2]

**1** Plant, Soil and Microbial Science Department, Michigan State University, East Lansing, MI, United States of America, **2** Great Lakes Bioenergy Research Center, Michigan State University, East Lansing, MI, United States of America

\* benucci@msu.edu

**Data Availability Statement:** NGS sequences are available on NCBI SRA database (BioProject ID: PRJNA421478; Sequence Read Archive Accession: SRP126404 ). Sanger sequences are available on GenBank with accessions from MH027189 to

## Abstract

Soil archives are an important resource in agronomic and ecosystem sciences. If microbial communities could be reconstructed from archived soil DNA, as prehistoric plant communities are reconstructed via pollen data, soil archive resources would assume even greater value for reconstructing land-use history, forensic science, and biosphere modelling. Yet, the effects of long-term soil archival on the preservation of microbial DNA is still largely unknown. To address this, we assessed the capacity of high-throughput sequencing (Illumina MiSeq) of ITS (internal transcribed spacer) and prokaryotic 16S rRNA genes for reconstructing soil microbial communities across a 20 years time-series. We studied air-dried soil archives and fresh soil samples taken from *Populus* bioenergy and deciduous forest research plots at the Kellogg Biological Station. Habitat and archival time explained significant amounts of variation in soil microbial α- and β-diversity both in fungal and prokaryotic communities. We found that microbial richness, diversity, and abundance generally decreased with storage time, but varied between habitat and taxonomic groups. The high relative abundance of ectomycorrhizal species including *Hebeloma* and *Cortinarius* detected in older soil archives raises questions regarding traits such as long-term persistence and viability of ectomycorrhizal propagules in soils, with relevance to forest health and ecosystem succession. *Talaromyces*, *Paecilomyces* and *Epicoccum* spp. were detected in fresh and across 20-year-old archived soils and were also cultured from these soils demonstrating their long-term spore viability. In summary, we found that microbial DNA in air-dried soils archived over the past 20 years degraded with time, in a manner that differed between soil types and phylogenetic groups of microbes.

## Introduction

Soil is a complex and diverse milieu consisting of a solid phase of minerals, organic matter, and microorganisms, as well as a porous matrix filled with gases and water [1]. Microbial diversity in soils is extreme, with a single gram harboring up to $10^{10}$ microorganisms and an

MH027207. S1-S16 Figs and S1 Table (S1 File), R scripts (S2 File), ITS and 16S otu_table.biom files (S3 and S4 Files), metadata files (S5 and S6 Files), representative sequences of fungal culture isolates (S7 File) are also provided as Supporting Information files and are available on GitHub (https://github.com/Gian77/Scientific-Papers-R-Code).

**Funding:** This research was supported through AgBioResearch NIFA, projects MICL02416 and MICL08541, and the Great Lakes Bioenergy Research Center, U.S. Department of Energy, Office of Science, Office of Biological and Environmental Research, under award number DE-SC0018409.

**Competing interests:** The authors have declared that no competing interests exist.

estimated diversity of between 4000–50,000 species per $cm^3$ [2]. Microbial ecologists have uncovered vast microbial diversity in soils over the past decade compared to what had been achieved in decades of classical sterile-culturing techniques, owing to advances in DNA-based high-throughput sequencing technologies and bioinformatic computation.

Fungi and prokaryotes play key roles in soil ecosystems, carrying out diverse biogeochemical processes which are essential to long-term (i.e. pedogenesis) and short term (i.e. soil functioning) ecosystem development. Some microbes are able to resist adverse environmental conditions, such as high temperature or desiccation, by entering in resting and reversible dormant states, or by forming dormant spore and spore-like structures [3,4]. Furthermore, intrinsic characteristics (e.g. size, composition) of organo-mineral complexes of soil aggregates can create advantageous conditions for the accumulation and preservation of nucleic acids in the soil environment, or they can be detrimental to its preservation [5]. Well preserved microbial DNA has been successfully extracted in permafrost samples of 400,000–600,000 years old [6].

Long-term soil preservation and archiving is essential to ecosystem ecology, as it allows the possibility to reanalyze samples and make direct comparison across studies and investigations into different aspects of the microbial communities across time [7]. Soil achieving was initiated by soil chemists, who used air drying and sieving as preservation treatments. Nowadays, several institutions worldwide curate collections of archived soils, which include the long-term soil archives of Rothamsted and Wageningen established in 1846 and 1879, respectively [8]. The limits to using soil archives to address questions in microbial ecology are still not well defined.

Previous studies have used soil archives to address specific research questions. For example, microbial communities in soils treated with animal manure differed from soil treated with chemical fertilizers, even after more than 50 years of dry storage [9]. Still, microbial communities and their metabolic activities may not be preserved at the same quality in archived soils [9]. In fact, functional attributes of soils change as the environment changes. For example, the ratio of soil enzymatic activity was found to be more significantly affected by freezing than by drying, especially in soil with high C (carbon) content, when compared to fresh soil [10,11]. A reduction in prokaryotic diversity was recently found in soil archives of the Central Museum of Soil Science in Russia, which had been air-dried and stored for over 70 years, relative to prokaryotic diversity in fresh soils [12]. Over short-term durations, Lauber et al. in 2010 [13] used high-throughput pyrosequencing of prokaryotic 16S rRNA to show that structure and diversity of prokaryotes in dried, frozen or soils stored at room temperature were not significantly different from fresh soil samples [13]. However, using the same technology and experimental settings but with a wider set of samples, Rubin and colleagues in 2013 [14] found that storage time and temperature do affect prokaryotic community composition and structure.

Only a few studies have investigated the impacts of soil storage conditions and time on eukaryotic communities. In one study, fungi and protists were detected in 18S rRNA gene libraries from 1975 air-dried archived soils [15]. The authors showed that nearly all taxa of eukaryotic soil microbes could be identified (fungi, cercozoans, ciliates, xanthophytes, heteroloboseans, amoebozoans, etc.), demonstrating that it is possible to study eukaryotic microbiota in samples from soil archives that have been stored for more than 30 years at room temperature. However, fresh soils were not available for comparison.

Microscopy-based spore assays have been used to investigate the diversity arbuscular mycorrhizal (AM) fungi in relation to $NO_x$ deposition in soil archives collected between 1937–1999 [16]. Soil nitrogen enrichment was shown to correlate with decreased productivity (spore biovolume) replacement of a formerly diverse AM community (richness = 29 species) with one composed of only seven taxa. Moreover, species of the larger spored genera *Acaulospora*, *Scutellospora* and *Gigaspora* and many *Glomus* species disappeared completely from the community over time.

Although several studies have tested the effect of different soil preservation techniques and storage times, none have investigated changes in prokaryotic and fungal community diversity and structure in a standard set of samples. In this set of studies, we used soils archived at the Long-Term Ecological Research (LTER) Network's Kellogg Biological Station (KBS) and next generation sequencing of 16S and ITS DNA markers to assess the capacity for preserving microbial DNA in air-dried soil archives over storage time. In doing so we identified fungal and prokaryotic taxa that are most susceptible and resilient to soil archiving. We were also able to model microbial OTU richness decay over time. Through this research we addressed the following questions: 1) How are fungal and prokaryotic diversity impacted by soil archival time and 2) how much variation is explained by soil storage time? 3) Which fungal and prokaryotic taxa are most resilient and sensitive to soil archiving? 4) Can molecular profiles of soil archives be useful in reconstructing land use history and management?

## Material and methods

### Study site

Archived soil samples were obtained from the Long-Term Ecological Networks (LTER) Kellogg Biological Station (KBS) *Populus* short-rotation (PS) woody biomass production sites and experimental deciduous forest (DF) sites. The DF habitat consisted of a canopy dominated by *Quercus rubra* L., *Pinus strobus* L., *Carya glabra* (Miller) Sweet, *Pinus resinosa* Aiton, *Prunus serotina* Ehrh., *Quercus alba* L. and *Picea abies* (L.) Karsten. The PS site was planted In 1989, using a hybrid poplar (*Populus deltoides* × *P. nigra*, also known as *Populus* × *euramericana* cv. Eugenei). Trees were planted as 15 cm stem cuttings on a $1 \times 2$ m row spacing, with nitrogen fertilizer applied only in the establishment year (123 kg N ha$^{-1}$). A cover crop *Festuca rubra* L. was planted in 1990 to help control soil erosion. Trees were harvested in 1999 and allowed to coppice (regrow from cut stems) for 10 years. Trees were harvested again in 2008. In May 2009, *Populus nigra* × *P. maximowiczii* cv. NM6 were planted as stem cuttings. Weeds were controlled with herbicides during the first 2 years of establishment and fertilizer was applied once (156 kg N ha $-^1$).

## Experimental design and soil sampling

Three experiments were designed to test the impact of sampling, plot and date on microbial reconstructions.

In experiment 1 we investigated the amount of variation in microbial community diversity and structure with increasing time of soil storage. We obtained air-dried soils from the LTER-KBS soil archive originating from 3 *Populus* short-rotation (PS) and 3 deciduous forest (DF) sites of the following years: 1995 (only for PS), 2000, 2005, 2010, 2014. In the Fall 2015, we collected 10 soil samples from the same selected sites for both PS and DF and mixed them together in site-composite samples according to the same methodology used in the KBS soil archive. Eighteen and 15 soils were analyzed in total for the PS and the DF forest sites, respectively.

In experiment 2 we tested the impact of sampling, DNA extraction, and PCR library construction on the reproducibility of results. To do this, 3 sub-samples (pseudo-replicates) of a single LTER-KBS soil (DF1) were sampled from each of the years (2015, 2010, 2005, 2000) to test reproducibility and consistency of the DNA-based approach used. For this experiment a total of twelve soil samples were analyzed.

In the third experiment we assessed local spatial variation in microbial community diversity and structure across select PS and DF sites at a single date, July 15, 2016. For this experiment

we sampled 3 soil samples each for 3 different PS and DF sites. In total we analyzed 18 soil samples.

## DNA extraction, amplification and Illumina library preparation

Soil DNA was extracted with the MagAttack PowerSoil DNA Kit (Qiagen, USA) on a King-Fisher Flex system (Thermo Fisher Scientific, USA). Fungal internal transcribed region (ITS) of the ribosomal RNA (rRNA) was amplified using ITS1F-ITS4 primers [17,18]. Prokaryotic V4 region of the 16S rRNA was amplified using the 515F-806R primers [19]. Amplicons libraries were prepared according to previous studies [20–22] Libraries were then sequenced on a MiSeq platform (Illumina Inc., USA) using the v3 kit 300PE. Complete submission of sequence reads can be found at NCBI SRA (https://www.ncbi.nlm.nih.gov/sra) archive [23] with the accession number SRP126404.

## Bioinformatic pipeline

Raw 16S reads were merged using PEAR [24], while for ITS only the forward read (ITS1) was used in the downstream analysis. Reads were demultiplexed according to the barcode index with QIIME [25]. Illumina adapters and sequencing primers were removed with cutadapt [26]. Sequences were then quality filtered, trimmed to remove conserved 18S and 5.8S motifs and to equal length [27,28], de-replicated, and clustered into operational taxonomic units (OTUs) based on 97% similarity using the UPARSE algorithm pipeline [29]. Singleton sequences were removed before clustering. Taxonomy assignments were performed in QIIME with the RDP Naïve Bayesian Classifier [30] Release 2.11 for 16S rRNA representative sequences, and with CONSTAX [31] for the fungal ITS. The Greengenes database [32] version gg_13_8 and UNITE database [33] version 7.1 2016-08-22 were used as 16S rRNA and ITS nrDNA taxonomic references, respectively.

## Statistical analysis

For each marker gene (i.e. ITS and 16S), *otu_table.biom* [34] with taxonomic classifications and metadata files, were imported into the R statistical environment [35] with the *phyloseq* package [36]. Contaminant OTUs sequenced from DNA extraction and PCR negative controls (no DNA added) were filtered from OTU tables. This accounted for 3 fungal OTUs (1.28% relative abundance) and for 16 prokaryotic OTUS (1.11% relative abundance). We filtered for potential PCR amplification and tag mismatching errors by removing OTUs containing <10 reads across all samples [37,38]. We then calculated rarefaction curves in *vegan* with the function "rarecurve" [39]. Given that the three different experiments showed different minimum library sizes, and were independent from one another, we first divided the dataset into 3 subsets, each representing the different experiments described above, and then normalized sample counts with the "rarefy_even_depth" function in *phyloseq*. All analyses and comparative metrics were calculated within each experiment, never between experiments. To test for the relationship between species richness and time (archived years) we compared linear, polynomial, poisson, and negative binomial models on datasets form Exp1 and Exp2. The best model was identified based upon the modified Akaike information (AICc) criterion. We used the function "simulateResiduals" in the *DHARMa* R package to create readily interpretable scaled (quantile) residuals for fitted (generalized) linear mixed models. Models were evaluated through implementations of ANOVA in both the *stats* and *car* packages. Coefficients' significance of the best models was assessed using the function "summary" for the linear and "sum" for the negative binomial models implemented in *stats* and *jtools*, respectively.

Multivariate analysis of β-diversity were assessed through non-metric multidimensional scaling (NMDS) unconstrained ordination analyses, to explore changes in community structures and implemented in the "metaMDS" function in *vegan* [39]. And also, by a Canonical Analysis of Principal Coordinates (CAP) constrained ordination analyses, to display differences in community structure related to the specific sample groups, implemented in the "capscale" function in *vegan*. CAP models were validated using ANOVA ($p \leq 0.05$). From the significant models the most parsimonious models were extracted using the "ordistep" function (perm. = 999) in *vegan*. All ordination analyses were performed on Bray-Curtis dissimilarity distance matrices [40]. A permutational multivariate analysis of variance (PERMANOVA) was used to test differences among a priori defined sample groups [41] with the functions "adonis" in *vegan* [39]. The replicate plot (i.e. site) was used as random effect in the models. Multivariate homogeneity of groups dispersions [42] was adopted to test for variance heterogeneity of a priori defined groups (β-diversity function) using the function "betadisper" in *vegan* [39]. To better detect taxa that showed variation across years of storage time, we first rescaled the read number of each OTU to 0–1 and then plotted in colored heatmaps. Putting OTUs on the same scale removes the differences in sequencing depth caused by differing library sizes between taxa (See Weiss et al. [43]). All graphs were plotted using *ggplot2* [44] and *graphics* [45] R packages. Putative mycorrhizal, pathogenic, and endophytic fungi were extracted from the ITS dataset using FUNGuild [46] and analyzed independently. fungal guild assignments were corrected manually according to available literature. A taxon-group association point-biserial correlation coefficient "r" analysis was used to assess the degree of preference and significance of taxa for years of soil storage with the function "multipatt" in the indicspecies R package [47]. To assess the dynamics of the microbial communities over time we calculated the turnover [*OTU gained + OTU lost / OTU in both timepoints*] using the "turnover" function in the *codyn* R package [48]. We then calculated the rate of community change as Euclidean distances of pair-wise communities across all years [49] to assess the direction of the observed community changes with the function "rate_change" in *codyn*.

## Soil viability

To determine whether microbial propagules in archived soils were still viable after archival, viability tests were carried out using 100 mg of DF and PS soils archived between 1995 and 2015. Given that the majority of soil bacteria have never been cultured [50], we focused on fungi. To culture fungi, soils were placed into 1 mL microcentrifuge tubes and were rehydrated with filter-sterilized deionized water. Each hydrated soil was briefly vortexed and allowed to rest at room temperature for three hours. Soil suspensions were then gently vortexed, and then 100 μL of suspension was pipetted and evenly spread onto 4 different agar media: potato dextrose agar (PDA) with and without antibiotics (chloramphenicol 100 mg/L and streptomycin 100 mg/L), Modified Melin-Norkrans (MMN) with antibiotics (chloramphenicol 100 mg/L and streptomycin 100 mg/L), Rose Bengal agar (RBA) with antibiotics (ampicillin 50 mg/L, streptomycin 100 mg/L, kanamycin 50 mg/L). Plates were incubated at 24°C and inspected daily. Each colony that grew was marked on the Petri dish and counted irrespective of their morphology. Morphological traits including colony color and hyphal characteristics (e.g. septa presence, clamp connections presence, thickness, branch angles) were used to distinguish morphologically unique colonies, which were transferred onto new media. To identify the taxonomy of these isolates, DNA from the fungal colonies developed in the oldest DF and PS soils was amplified and sequenced on a Sanger sequencer (Applied Biosystems, USA) as previously described [51]. Sequences obtained were accessioned to NCBI sequence archive with accession numbers from MH027189 to MH027207 and available in S7 File.

# Results

## Amplicon sequencing

A total of 1,616,765 (average 26,946 ± 9,135 per sample) fungal ITS and 3,332,238 (52,893 ± 24,300 per sample) prokaryotic 16S raw reads resulting from 59 samples (plus 3 controls) were analyzed, respectively. After filtering according to read quality and sample controls, 1,541,011 fungal ITS and 2,640,408 prokaryotic 16S reads remained for community analysis. After removing non-target OTUs and control contaminants we remained with a total of 3030 fungal and 6760 prokaryotic OTUs. Graphical representations of the overall distribution of sample libraries, and distribution of sample libraries divided by year of storage, are provided in S1 and S2 Figs in S1 File. Both the distribution of sample libraries as well as rarefaction curves (S3 Fig in S1 File) of the fungal and bacterial datasets benefited from normalization (rarefaction to minimum library depth) after separating the data from the different experiments. Datasets were divided in 3 sub-datasets according to the experimental design, as reported in the Material and Methods section and rarefied to minimum sequencing depth. The dataset of experiment 1 (hereafter Exp1) included 2289 (2148, normalized) fungal and 6524 (5415) prokaryotic OTUs, experiment 2 (Exp2) with 980 (919) fungal and 4421 (3483) prokaryotic OTUs, and experiment 3 (Exp3) with 2312 (2218) fungal and 6484 (6319) prokaryotic OTUs.

## Alpha diversity

Boxplots showing fungal (Fig 1A) and prokaryotic (Fig 1B) rarefied richness were plotted for Exp1, Exp2 and Exp3. For the same datasets, boxplots showing values for observed richness and Shannon diversity index are available in Supplemental Material for both fungi (S4A and S5A Figs in S1 File) and prokaryotes (S4B and S5B Figs in S1 File), respectively. In DF and PS soils (Exp1), a trend of decreased species richness with archiving time was evident for both fungi (Fig 1A Exp1) and prokaryotes (Fig 1B Exp1). The same pattern of diversity decay with time is evident when only soils collected in site DF1 (Exp2) are included in the analysis (Fig 1A and 1B). This pattern was strongest for fungal rarefied and observed richness, as well as for Shannon diversity index, and was less pronounced for prokaryotes (S4 and S5 Figs in S1 File). Variation in fungal and prokaryotic rarefied, observed, and Shannon diversity index in fresh soils (Exp3) was also detected among different sites and according to different habitats (Fig 1A Exp3, S4 Exp3 and S5 Exp3 Figs in S1 File).

The models that performed the best at describing the relationship between storage time and rarefied species richness, selected among all the models were fit to explore the richness/storage time relationship (S6-S11 Figs in S1 File), were negative binomial models for the fungi (both in DF and in PS habitats), a quadratic polynomial model for the Prokaryotes in the DF habitat and a logarithmic model for the Prokaryotes in the PS habitat (Fig 2).

In all the models, except on the model for prokaryote DF habitat, rarefied species richness decreased significantly with storage time consistent with the hypothesis that increased storage time negatively affects community alpha diversity. This was particularly evident for fungi where negative binomial models fit the data at best. Storage time explained 56% in the DF and 52% in PS of the decrease in fungal rarefied richness (Table 1, Fig 2, S6-S8 Figs in S1 File). Similar trends were observed for prokaryotic rarefied richness but only in PS, where a logarithmic model fit the data at best and showed that storage time explained 55% decrease in rarefied species richness. In DF, a quadratic model was the best fit for prokaryotes. The 59% of rarefied richness was explained by variation in storage time (Table 1, Fig 2, S9-S11 Figs in S1 File).

The results obtained for Exp1 in the DF habitat were supported by the models obtained from the Exp2 dataset (DF1 site only). In this case, the best models to fit our data distribution

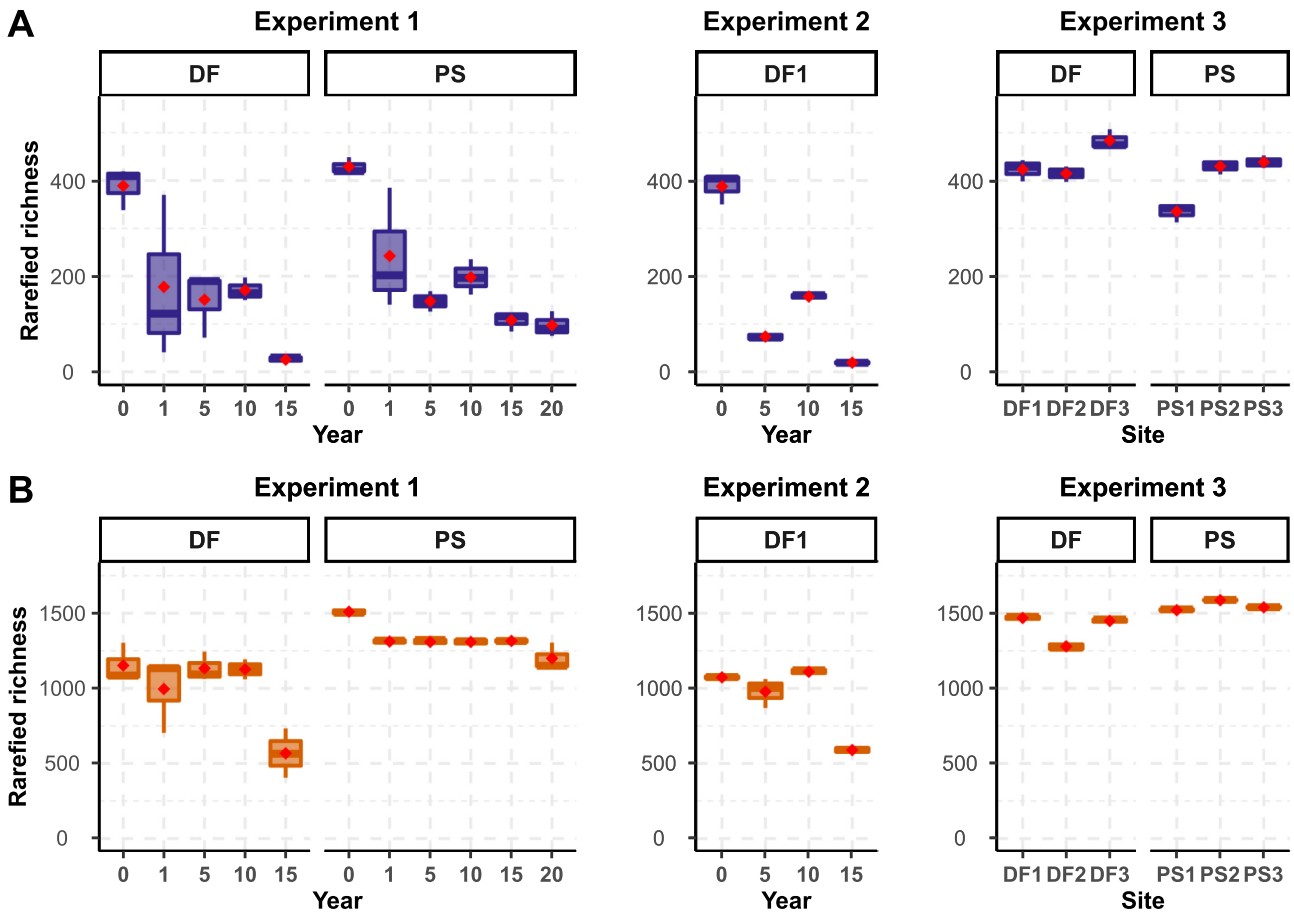

**Fig 1.** Fungal (A) and prokaryotic (B) rarefied richness boxplots (n = 3) in Experiment 1, Experiment 2, and Experiment 3 (See M&M for details). Plots are faceted by DF (deciduous forest) and PS (*Populus* stand) habitats. Red diamonds represent the mean of the distribution.

were negative binomial and logarithmic for the fungi (S7 Fig in S1 File) and polynomial quadratic or cubic for the prokaryotes (S10 Fig in S1 File).

## Beta diversity

Major differences in community structure based upon deciduous forest (DF) and *Populus* stand (PS) habitat and time (year) of storage were found. Fungal (Fig 3A) and prokaryotic (Fig 3B) NMDS ordinations, for both fungi and prokaryotes, showed that samples cluster in two main groups according habitat along the first axis (i.e. NMDS1) and according year along the second axis (i.e. NMDS2), respectively. High non-metric and linear $R^2$ of observed dissimilarities against ordination distances demonstrated a good representation in the ordination space (S12 Fig in S1 File). Fungal (Fig 3C) and prokaryotic (Fig 3D) canonical analysis of principal coordinates (CAP) ordinations showed similar clustering patterns to NMDS. The most parsimonious CAP model (site + year + habitat) for Fungi explained about 27% of the community variance and was significant ($F_{6,24}$ = 2.823, p = 0.001 after Bonferroni correction and 999 perm.) The most parsimonious model (site + year + habitat + year:habitat) for the prokaryotic community was statistically significant ($F_{10,22}$ = 3.602, p = 0.001 after Bonferroni correction and 999 perm.) and explained about 45% of the variance (Fig 3, Table 2).

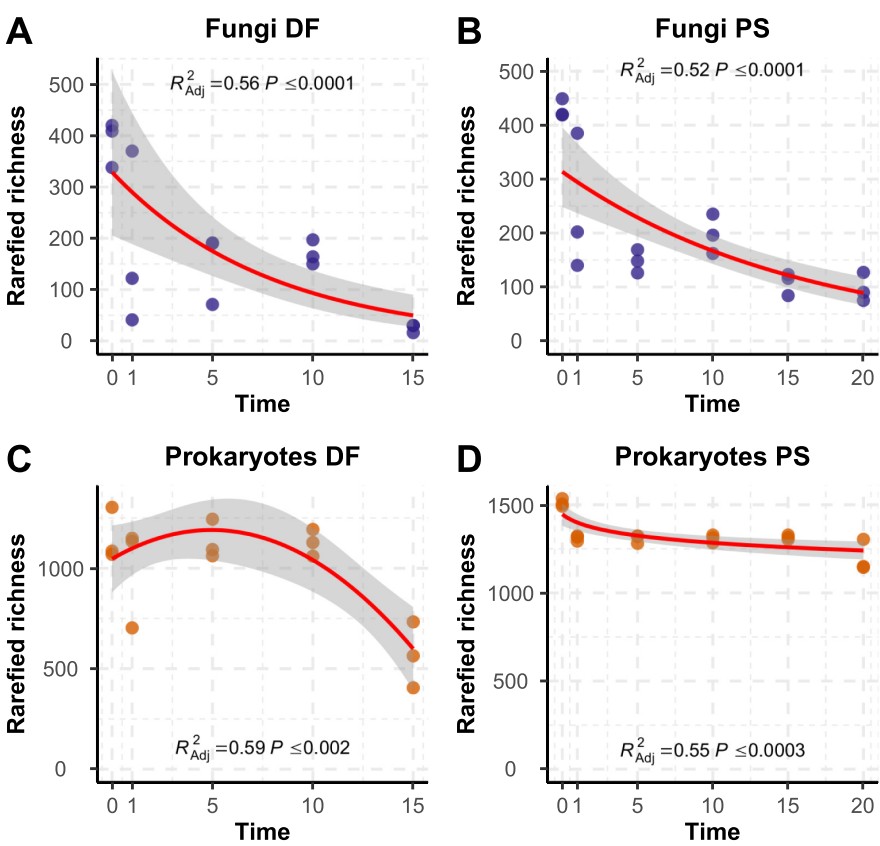

**Fig 2. Best models of rarefied richness variation according to storage time (Exp1).** Points represent samples of fungal (A) and prokaryotic communities (C) in DF (deciduous forest) habitat and fungal (B) and prokaryotic (D) communities in PS (*Populus* stand) habitat.

PERMANOVA (function "adonis") analysis using the same models also showed significant differences between *a priori* selected groups (Table 2). A significant interaction between storage time and sample origin (i.e. habitat) was detected in the prokaryotic communities that

**Table 1. OTU richness and storage time model details.**

| Fungi | PS | Fungi | DF |
|---|---|---|---|
| equation | y = 313.68 + 0.94x$^2$/9.63 | equation | y = 328.85 + 0.88x$^2$/2.60 |
| Chisq | 34.34 | Chisq | 14.82 |
| R$^2$ | 0.59 | R$^2$ | 0.54 |
| Adj. R$^2$ | 0.56 | Adj. R$^2$ | 0.51 |
| p-value | <0.0001 | p-value | <0.0001 |
| **Prokaryotes** | **PS** | **Prokaryotes** | **DF** |
| equation | y = 1445.94–67 log(x+1) | equation | y = 1046.42 + 58.02x – 5.86x$^2$ |
| F$_{1,16}$ | 21.91 | F$_{2,12}$ | 10.995 |
| R$^2$ | 0.58 | R$^2$ | 0.65 |
| Adj. R$^2$ | 0.55 | Adj. R$^2$ | 0.59 |
| p-value | <0.001 | p-value | <0.01 |

Model equations as well as Chisq, R$^2$, adjusted R$^2$, F and p-values of the best models are reported. Models were calculated on Experiment 1 data for both fungal and prokaryotic communities.

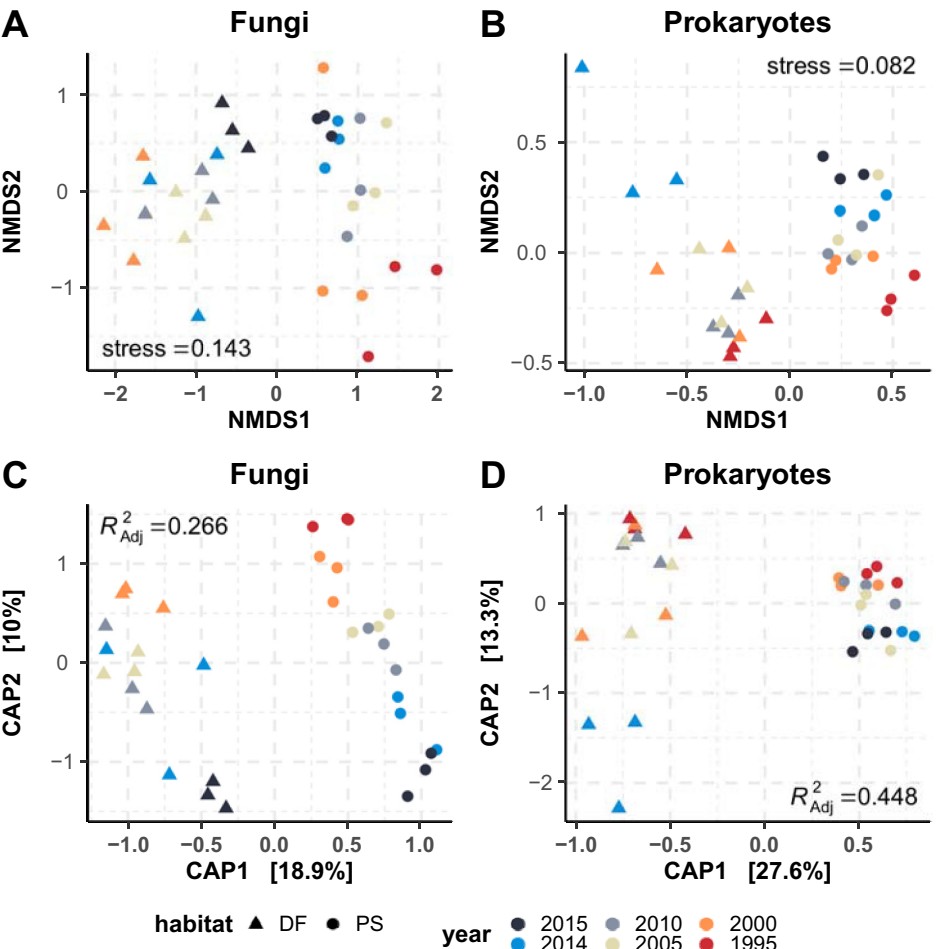

**Fig 3. Non-Metric Multidimensional Scaling (NMDS) and Canonical Analysis Of Principal Coordinates (CAP) ordinations.** Plots for fungal (A, C) and (B, D) prokaryotic communities are shown. Analyses were performed on even-depth normalized read abundance and bray-curtis distance.

**Table 2. Canonical Analysis of Principal Coordinates (CAP) and Permutational Multivariate Analysis of Variance (PERMANOVA) results.**

|  | CAP | | | PERMANOVA | | | |
|---|---|---|---|---|---|---|---|
| **Factor** | **Df** | **Adj.R$^2$** | **P-value** | **Df** | **F-value** | **R$^2$** | **P-value** |
| **Fungi** | | | | | | | |
| **block + year + habitat** | 6 | 0.266 | 0.001 | | | | |
| year | 5 | 0.113 | 0.001 | 5 | 2.072 | 0.221 | 0.004 |
| habitat | 1 | 0.182 | 0.001 | 1 | 7.719 | 0.165 | 0.004 |
| **Total** | 32 | | | 32 | | | |
| **Prokaryotes** | **Df** | **Adj.R$^2$** | **P-value** | **Df** | **F-value** | **R$^2$** | **P-value** |
| **block + year + habitat + year:habitat** | 10 | 0.448 | 0.001 | | | | |
| year:habitat | 4 | 0.065 | 0.001 | 4 | 1.918 | 0.122 | 0.024 |
| **Total** | 32 | | | | | | |

Data refers to the Experiment 1 dataset. The models adopted were the most parsimonious.

explained about 6.5% and 12% of the variance detected in the CAP model and PERMANOVA $R^2$, respectively.

The homogeneity of group dispersion analysis (function "betadisper") showed significant differences of group variances across different years of soil storage in the fungal ($F_{5,27}$ = 3.13, p = 0.023, after Bonferroni correction and 999 perm.) and prokaryotic communities ($F_{5,27}$ = 4.29, p = 0.005, after Bonferroni correction and 999 perm.), respectively, but no significant differences between the sample's habitat (PS or DF) both for fungi and prokaryotes (S13 Fig in S1 File). Significant effect of the sampling site on fungal and community diversity was also detected, as shown in the CAP performed on soils DNA replicates (Exp2) and those collected fresh in 2016 dataset (Exp3) shown in S14 Fig in S1 File. Ordinations showed a high consistency in DNA replicates of the same year with samples clustering on top of each other in Exp2 and difference in fresh soils (Exp3) primarily by habitat (CAP1) and then by site (CAP2).

## Heatmap analysis

Class level heatmaps of fungal ITS based on the Exp1 dataset (Fig 4) showed that the biggest drop in richness and abundance happened after 1 year, with most of the fungal taxa undetectable after 10 years of storage time. Interestingly, relative abundance and OTU richness did not always consistently decrease with storage time (Fig 4). This was true for both DF and PS soils. For example, while the OTU abundance and richness of Agaricomycetes and Glomeromycetes decreased with storage time in DF soils, it increased after 10 years in PS soils, and then decreased again at 15 and 20 years. Additionally, the presence/absence and abundance pattern of taxa such as Wallemiomycetes and Olpidiomycetes remained constant across all years of soil storage. To better detect abundance-change patterns in the DF and PS sites, we separated the fungal genera according to functional guilds. These guild data were visualized individually with a heatmap that shows abundance-change patterns for mycorrhizal, pathogenic, and endophytic fungi (Fig 5). Richness and relative abundance of ectomycorrhizal species such as *Tomentella*, *Inocybe*, *Cortinarius*, *Hebeloma* and *Glomus* varied in the PS across sampled years. Specifically, these taxa were present and abundant in years before and after 2008 and 1999 (soils between 5 and 15 years of storage), the year when the trees were harvested, and a new plantation was established. *Tomentella*, *Inocybe* and *Russula* showed a similar pattern in the DF soils, with peaks in abundance and richness at 0 and 10 years, but no detection after 10 years. Pathogenic fungi in the PS habitat appeared to have a shorter persistence in the stored soils compared to mycorrhizal and endophytic fungi. After 5 years of storage few pathogenic fungal genera were detectable, with the exception of *Fusarium*, which was detected until immediately after the *Populus* harvest. *Fusarium* was more abundant and OTU-rich in the PS soils. Regarding endophytic fungi, *Mortierella* was the most abundant in both habitats, particularly within years 0 and 10, underlining the survival ability to desiccation and the high phylogenetic diversity within the genus.

Richness and relative abundance of prokaryotic taxa also appeared to be affected by storage time and habitat as shown in the heatmap of S15 Fig in S1 File. Planctomycetia and Alphaproteobacteria had the highest richness across the different habitats. All taxa decreased with storage time, except Bacilli that showed the opposite trend in both habitats. Aside from this group, no clear pattern or decrease/increase or abundance was found for prokaryotes in relation to the harvest of *Populus*.

## OTU turnover

A higher total turnover of OTUs (~ 75%) between time points was found for fungi compared to prokaryotes (~50%) (Fig 6A). Additionally, we detected a similar fluctuation pattern in

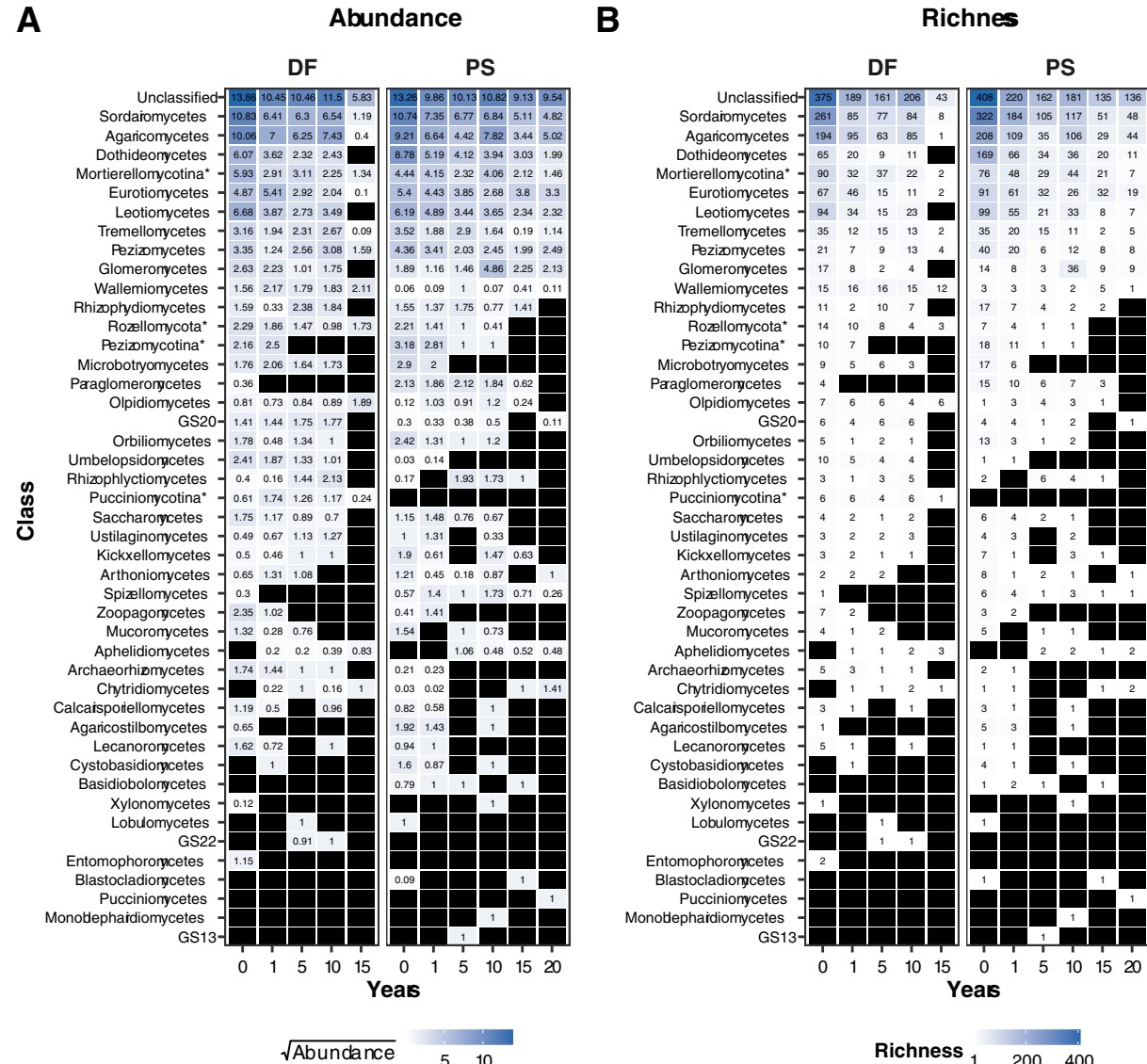

**Fig 4. Heatmap of relative abundance and OTU richness of all the fungal classes detected in DF (deciduous forest) and PS (*Populus* stand) soils after increasing storage years.** Taxon relative abundance was square root transformed to improved visibility. Black indicates zero relative abundance or absent taxa. For taxa marked with * subphylum was used instead of Class incertae sedis.

fungi DF and prokaryotes DF with a reduced OTU loss and increased gain after 15 years (between 2000 and 2005). In PS, the lower number of OTUs gained after 10 years (between 2005 and 2010) can be explained by the fact that trees were cut in 2008. The community compositional changes represented by variations in Euclidean distance indicate the presence of evident directional changes in DF (i.e. the communities are increasingly dissimilar over time), both in fungal and in prokaryotic communities, compared to PS habitat where change in species richness and turnover are less dependent to storage time (Fig 6B).

## Fungal species that persist in archived soils

In total, we obtained 16 and 100 unique fungal isolates from DF soils and PS soils, respectively. Both the number and diversity of fungal isolates declined precipitously with soil age. The

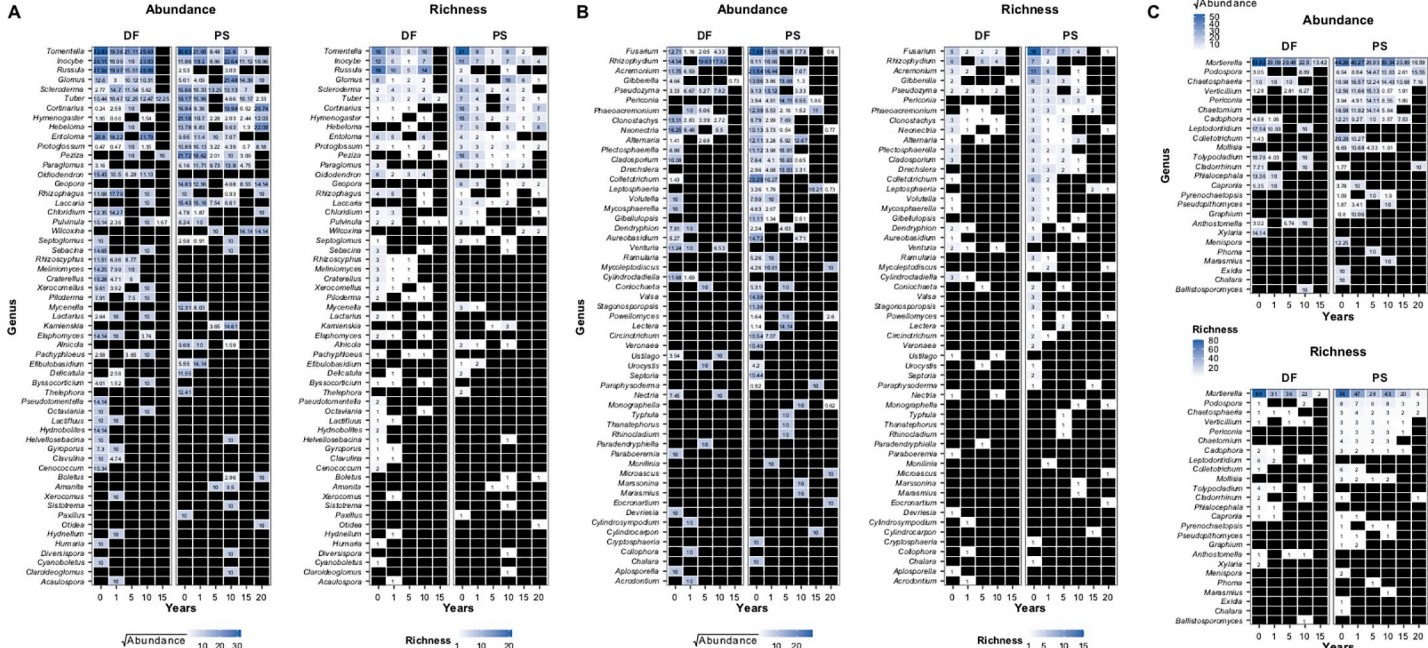

**Fig 5. Heatmap of relative abundance and OTU richness of only the mycorrhizal (A), pathogenic (B), and endophytic (C) fungal genera after increasing storage years.** Plots are faceted by DF (deciduous forest) and PS (*Populus* stand) soils. Taxon relative abundance was square root transformed to improved visibility. Black indicates zero relative abundance or absent taxa.

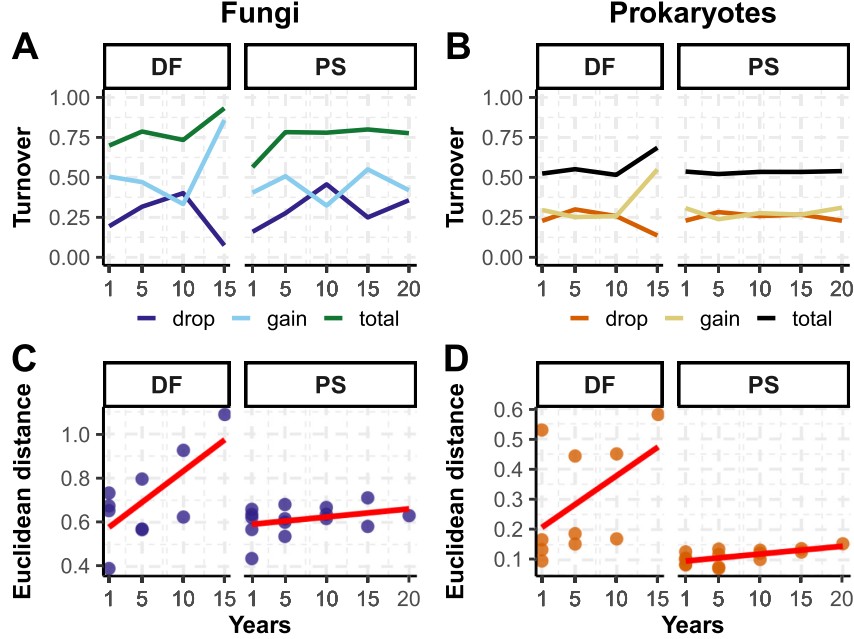

**Fig 6. Taxonomic turnover and compositional changes in fungal and prokaryotic communities.** Total OTU turnover, OTU gained and OTU dropped, for fungal (A) and prokaryotic (B) communities in both DF (deciduous forest) and PS (*Pupulus* stand), and compositional community changes described by Euclidean distances in fungal (C) and prokaryotic (D) communities in DF and PS.

oldest soil from which fungal colonies emerged was DF1 and it was 10 years old (i.e. 2005). Isolates from 20-year-old (i.e. 1995) soils were obtained from PS1, PS2, and PS3 (S1 Table in S1 File). ITS rDNA sequences derived from cultured isolate matched with sequences obtained from Illumina ITS rDNA sequencing from these soils, which we classified to *Talaromyces*, *Auxarthron*, *Epicoccum*, *Paecilomyces*, in order of decreasing total abundance (S16 Fig and S1 Table in S1 File).

## Discussion

Long-term soil archives are maintained by research institutes globally, but their utility for studying and reconstructing soil microbiomes remains unknown. In this study, we set out to determine how well soil archives preserve fungal and prokaryotic DNA, and the legacy that it may contain. Further, we sought to identify those taxa that were the most resilient and sensitive soil archival. Our results demonstrate that soil samples preserved by air-drying and long-time soil archival can be used to identify differences in microbial communities along different temporal, spatial and organizational scales, as well as in response to soil traits and management [9,12,52]. However, we did find that fungal and prokaryotic diversity both decrease with time, and that this response differed between taxa and study sites.

Although long-term storage of air-dried soils is known to generally reduce microbial diversity [12,15,53], effects of increased archival time on microbial communities or specific microbes in archived soils is less explored. Our results demonstrate that fungal, and to a lesser extent prokaryotic, OTU richness and diversity in soils from a *Populus* short-rotation crop (PS) and deciduous forest (DF) decrease significantly ($P \leq 0.01$) and with time of soil archiving (Figs 1 and 2, S2-S7 Figs in S1 File, Table 1). The most precipitous drop in richness occurs between 0 and 1 after sampling (Figs 1 and 2, S2-S7 Figs in S1 File), particularly for the fungi, while changes in later years seem to have more erratic behavior (going up and down after 5, 10 and 15 years). That large initial drop in richness indicates that it may be the drying rather than the age that is causal. Compared to spores, fungal mycelium may be more prone to autolysis as vesicles containing nucleases are ruptured. Thus, fungal DNA originating from mycelium may not archive as well as DNA from spores, and if this is the case, could help explain the drop in diversity that was observed after 1 year. The situation is less clear for the prokaryotes, but it seems that the drying process has less effect on prokaryotic cells compared to fungal ones. This may have to do with the proportion of cells that are actively growing compared to all the cells present. While the majority of bacterial taxa decreased with storage time, Bacilli increased constantly with archival time regardless of the soil habitat. This is likely due to their spore-forming ability but could also be explained by active growth during archival, lower resistance to mechanical rupture during DNA extraction years after years, or slower cell degradation. In general, resistant spores by definition have evolved to protect their intact DNA and cell viability and may be expected to naturally be more protected against drought (archival) than living hyphae, and less biased by DNA extraction. We did not directly measure DNA degradation, active vs inactive cells, nor DNA in spores vs. mycelium, but further experiments that manipulate factors could help to explain mechanisms that underlie soil archival DNA stability.

Interestingly, the magnitude of this storage effect on the microbial communities varied according to habitat where the soils were derived, as well as by year. The presence of a significant interaction between habitat and year confirmed that these two variables are not independent and support the hypothesis that the preservation of microbial community DNA may differ depending on specific soil characteristics and the specific environmental condition present at or during sampling and archival [5]. For instance, it has been shown that degradation of synthetic environmental DNA added to microcosm containing different natural soils varies

with moisture, temperature and habitat characteristics (e.g. organic carbon content) [54]. Real differences in community across time, such as through succession rather than differences from sample degradation with time, offer another possibility of how storage time and soil origin are not independent from each. Additionally, it is notable that site differences/time effects are disproportionate between microbes and show stronger alpha- and beta-diversity impacts on fungal compared to prokaryotic communities (Fig 3).

## Ectomycorrhizal taxa may be indicators of land use history

The use of microbial community DNA profiles to indirectly detect variation on associated vegetation is not new. Gellie and colleagues (2017) used high-throughput metabarcoding of environmental soil DNA as an effective tool to demonstrate the return of the native prokaryotic community in an old field following native plant revegetation [55]. Similarly, Clemmensen et al. (2015) used 454-pyrosequencing of the ITS fungal rDNA and showed that ectomycorrhizal fungal community patterns correlate with differences in C sequestration from root-associated mycelium during successional development of boreal forest [56]. In this study, we show that a few ectomycorrhizal species (e.g. *Tomentella*, *Inocybe*, *Cortinarius*, *Glomus*) were abundant before *Populus* trees were harvested, and dropped in abundance or became absent after harvests in 1999 (between 15 and 20 years of soil storage) and 2008 (between 5 and 10 years of storage), which appears to reflect the known land use history. Thus, these ectomycorrhizal species may be considered potential indicators of the presence of tree hosts. However, it is worth noting that other ectomycorrhizal taxa such as *Inocybe*, *Hymenogaster*, and *Protoglossum* were detected in nearly all the years and plots, regardless of the presence or not of *Populus* trees. Other taxa, including *Peziza*, seemed not to be affected by plant habitat but decreased progressively with storage time. These data demonstrate how some propagules of some fungal taxa can persist in archived soils for decades with their DNA sufficiently intact to be detected using NGS amplicon sequencing, and the challenges in interpreting NGS data from archived samples. It remains to be determined whether ectomycorrhizal fungal spores detected in older soil samples remain viable, but spore longevity would be an important trait to score for ectomycorrhizal (and other) taxa [57]. "Trap-plant" seedling bioassays could be used to bait for ectomycorrhizal and other biotrophic microbes, including endophytes and pathogens, in soil archives to better assess propagule viability across different trophic groups [51].

## Do soil archives have utility to future microbial ecologists?

Archived soil samples represent an important potential resource for microbial ecologists. Although not directly measured, our data suggests that DNA degradation and microbial activity (e.g. growth, production of spores) by xerophiles may comprise the integrity of air-dried soil archives for soil microbiome analysis. Therefore, identifying long-term storage protocols that are compatible with stabilizing soil microbiome DNA integrity are still needed. Ideally, approaches would be energy and time efficient. Soil archives represent a long-term investment in space and resources [9]. In molecular ecology, flash freezing soil samples and storing at –80˚C is considered best practice for preserving microbial nucleic acids prior to extraction. However, this practice is energy intensive. Alternative methods include freeze-drying, vacuum packing and storage at –20˚C, or perhaps storage in ethanol (for DNA) or DNA buffer [58]. Standardized soil archiving protocols should consider that storage techniques, procedures, conditions and different soil geochemical traits may produce artifacts that could be interpreted as microbial community shifts. In addition, organisms may respond differently to archival depending on environmental and soil variables. For example, Cui et al. (2014) showed that freezing changed the microbial communities less than air drying, and that the microbial

community they detected in agricultural soils was more stable than that in forest soils [59]. They also found fungal communities were more stable than prokaryotic communities, contrary to the results we present here. Ivanova et al. 2017 showed that long-term storage of soils in a museum exerted a greater impact on the microbiomes of podzolic soils, while chernozem soils preserved better in their native community [12].

Soil archiving for microbial ecology purposes may benefit from precise and strict standard operating procedures and standards that minimize microbiome changes in samples preserved across years and sites, which may vary due to environmental conditions (e.g. temperature, residual moisture, humidity, geochemical). In our study, PCR amplification was used as a technique to measure microbial community member presence and relative abundance, but this method is only semi-quantitative and limited since amplification can vary depending on template diversity and amount.

Thus, an increase in relative abundance over time could be due to the decline of template DNA in a sample over time, could be caused by microbial growth, or could be a result marker-gene and metagenomic sequencing measurements that are biased toward detecting some taxa over others. It is also known that amplicon sequencing approaches have embedded multiplicative biases (e.g. taxon extraction efficiency, selective PCR amplification, different copy number of ribosomal marker genes, bioinformatic analysis) that distort measurements and perception of real taxon abundances, which are impossible to estimate without the presence of internal controls that are sequenced alongside true samples [60]. For example, the increasing diversity of Agaricomycetes in the PS plots over time leads us to hypothesize that some fungi may be biologically active during archival. Future approaches that are PCR free, or include spike-in controls, may help to provide a more quantitative framework for addressing questions in microbial ecology.

In conclusion, this study demonstrates that in archived air-dried soils microbial DNA decay over time. We identify a number of fungal and bacterial taxa that are resistant to DNA decay, and others that are more sensitive. Moreover, significant interaction between storage time and sample origin were found indicating that the preservation of the original microbial community may vary according to specific soil physicochemical characteristics and site conditions on top of DNA degradation. We further demonstrate that fungal propagules (e.g. spores) of *Talaromyces*, *Auxarthron*, *Epicoccum*, *Paecilomyces* are able to persist, germinate, and grow even after 20 years of storage. Interestingly, these same fungi were detected at high relative abundance through metagenomic high-throughput sequencing, and share traits of being keretinolytic and in having asexual phases where spores are produced in abundance. The establishment of standard protocols for long-term microbiome DNA preservation from soil and environmental samples would serve future ecologists who may want to address microbial succession and microbiome turnover between energy crop harvest cycles, crop rotations or land-use changes. Towards this end, approaches to verify propagule viability, to identify relic DNA that remains intact in soils after cell death, and to provide absolute quantification of microbial communities will be valuable tools for the community [61].

## Supporting information

**S1 File.**
(PDF)

**S2 File.**
(R)

**S3 File.**
(BIOM)

**S4 File.**
(BIOM)

**S5 File.**
(TXT)

**S6 File.**
(TXT)

**S7 File.**
(FASTA)

# Acknowledgments

We are grateful to the GLBRC and KBS LTER Network for facilitating this study. In particular, we thank Stacey VanderWulp, Phil Robertson, Frances Trail and Kristy Gdanetz for assistance in obtaining archived soil samples and helpful discussions.

# Author Contributions

**Conceptualization:** Gregory Bonito.

**Data curation:** Gian Maria Niccolò Benucci, Gregory Bonito.

**Formal analysis:** Gian Maria Niccolò Benucci.

**Funding acquisition:** Gregory Bonito.

**Investigation:** Gian Maria Niccolò Benucci.

**Methodology:** Gian Maria Niccolò Benucci, Bryan Rennick.

**Project administration:** Gian Maria Niccolò Benucci, Gregory Bonito.

**Software:** Gian Maria Niccolò Benucci.

**Supervision:** Gian Maria Niccolò Benucci, Gregory Bonito.

**Validation:** Gian Maria Niccolò Benucci, Bryan Rennick.

**Visualization:** Gian Maria Niccolò Benucci.

**Writing – original draft:** Gian Maria Niccolò Benucci, Bryan Rennick, Gregory Bonito.

**Writing – review & editing:** Gian Maria Niccolò Benucci, Bryan Rennick, Gregory Bonito.

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
