## [Decision Letter · Decision Letter 0]

24 Jun 2020

PONE-D-20-15299

Patient propagules: Do soil archives preserve the legacy of fungal and prokaryotic communities?

PLOS ONE

Dear Dr. Benucci,

Thank you for submitting your manuscript to PLOS ONE. After careful consideration, we feel that it has merit but does not fully meet PLOS ONE’s publication criteria as it currently stands. Therefore, we invite you to submit a revised version of the manuscript that addresses the points raised during the review process.

With a few exceptions, the reviews were quite complimentary. Reviewer #1 asked for clarification on a few experimental details and, if available, the inclusion of soil chemistry data.  Reviewer #2's thoughtful comments recommend the incorporation of several discussion points and sensible revisions to data presented in the heat map. I concur with all the reviewers' points.

We look forward to receiving your revised manuscript.

Kind regards,

Daniel Cullen

Academic Editor

PLOS ONE

Journal Requirements:

Reviewers' comments:

Reviewer's Responses to Questions

**Comments to the Author**

1. Is the manuscript technically sound, and do the data support the conclusions?

Reviewer #1: Yes

Reviewer #2: Yes

2. Has the statistical analysis been performed appropriately and rigorously? 

Reviewer #1: Yes

Reviewer #2: Yes

3. Have the authors made all data underlying the findings in their manuscript fully available?

Reviewer #1: Yes

Reviewer #2: Yes

4. Is the manuscript presented in an intelligible fashion and written in standard English?

Reviewer #1: Yes

Reviewer #2: Yes

5. Review Comments to the Author

Reviewer #1: L46-49: I am not sure how this fits with the rest of the manuscript. As far as I can tell the work details degradation of DNA in soil preserved by the same method.

L164: Isn’t the kit the MagAttract?

L165: Capitalization of fungal

L166: Capitalization of prokaryotic.

L168: Period needed after the reference.

L194-198: Just to be clear, the samples were rarefied to the same depth once in phyloseq and

not again in vegan.

L208-209: Was the Bray-Curtis used as the distance?

L217: a prior needs to be italicized

L455-457: The idea of DNA degradation is due in part to environmental variation has been mentioned several times in the manuscript. I think it would be advantageous to include some data on soil physiochemistry to help support this point.

L463:469: Same comment here.

L473-474: Consider making this a statement rather than a question.

L501: What is meant by ‘microbial activity’?

L530-533: Same comments as for L455-457.

Reviewer #2: The study by Benucci is interesting and important, and I thought it was well conducted and analyzed. I do think that it is slightly over-interpreted in some parts and I discuss these below, but overall such problems are quite minor. The writing and visual presentation is clear and accessible, although I have a few suggestions for improvement primarily related to the heat-map figures. Overall, I thought it was an excellent study.

With the fungi it’s very clear that most of drop in richness occurs between year 0 and 1 (Fig 1 A & B). In fact, the somewhat erratic behavior (year 10, DF exp 2) looks like some of the changes in later years are primarily noise. That large initial drop in richness suggests that it's the drying not the age that is causal. I suspect that is because much of year 0 signal is from mycelium, which may be more prone to autolysis as vesicles containing nucleases are ruptured. Whatever the cause, it is a striking result that is not really mentioned. It could be emphasized more when methods for archiving are brought up in the discussion.

The culturing part of the experiment is pretty limited and not really well incorporated into the rest of the study. I suggest dropping it, as it adds very little. The weedy taxa found are those that are always found. I would be very surprised if others have not already reported these from archived soils, but no such studies are referenced. Dilution to end point culturing would be needed to see other taxa.

The heat maps are color-coded backwards – red (i.e, hot) is usually the highest value not the lowest. The highest intensity of red in these is counter-intuitively the least abundant/least rich, and the sea of white looks like missing data instead of intermediate values. I suggest a single gray scale going from white (0) to black (max), or white (0) to very dark red (max). The only reason for 2 colors in heat maps (e.g., gene expression data) is that usually they are referenced to increases (red) versus decreases (blue) from some starting point or control. Since there is no control or starting point used as a reference a single color would work better to communicate the patterns of change.

However abundance and richness are represented, interpreting changes in them overtime is not straight-forward, and the authors should be careful on how they do this. Since these are relative abundances, they can increase over time even if their absolute abundance has decreased; they simply have to decrease less than other taxa to show an increased in relative abundance. The increased presence of Agaricomycetes in figure 4, for example, could be caused by resistant spores that lose their intact DNA at slower rates than taxa with less resistant spores. Wallemiomycetes could have exactly the same explanation, alternatively maybe these xerophiles could grow in the dried soil, but the authors have certainly not provided evidence to support that here, in spite of their indirect claims made on line 501. I think one would need qPCR to resolve this.

Changes in richness of particular taxa is similarly not straight forward to interpret from these data. We can assume taxon richness is a property of the sample and that it can only go down with time (i.e., we can lose taxa, but not gain them). When it appears to go up, it means that our ability to detect richness of those taxon has changed. This may be due to increased abundance through growth, which seems unlikely but not impossible, or increased relative abundance due to loss of previously more abundant taxa. This is because as absolute quantity of DNA drops, more of the sequences obtained are derived from the remaining resistant pool.

Real differences in community across time rather than differences from sample degradation with time offer another way that time and site are not independent. The greater directional change in DF communities, might be based on this rather than higher DNA degradational changes that the authors seem to assume. In other words, the result may reflect underlying successional changes in DF, with PS less susceptible to these because it’s maintained in a narrower successional window by harvest. To tell the difference one would need archived DNA extractions. Then one could compare year 15 archived soils to year 15 archived DNA (extracted 15 years ago). This idea could be tied in the discussion of archive methods.

This idea of biological differences underlying some of the observed differences among years is discussed in the context of Tuber, Hebeloma and Cortinarius abundance changes that coincide with logging of the poplar sites, but looking at the unlogged DF patterns one can also see various EM taxa wane and wax without a logging event. This should caution the authors about cherry-picked examples, and probably says there is significant year to year variation even in undisturbed forest.

P 7 – methods make it sound like the whole ITS was sequenced for the fungi. Is that correct? It’s almost always ITS1 or 2 spacer because the entire region is too large. In any case the real primers used are probably not ITS1f and 4, because they have none of the adapters and barcodes needed. Please correct this.

lines 297 and 310 “at best” is used in a peculiar context – reword for clarity

6. PLOS authors have the option to publish the peer review history of their article (what does this mean?). If published, this will include your full peer review and any attached files.

Reviewer #1: No

Reviewer #2: Yes: Thomas D Bruns

---

## [Author Response · Author response to Decision Letter 0]

21 Jul 2020

Dear Editor, 

we would like to thank you and the reviewers for your insightful comments. Below (in red text) you will find our response to reviewer comments, in a point-by-point reply that includes having addressed the comment in the revised manuscript. Specifically, we have clarified experimental details requested by reviewer 1, and the incorporation of several discussion points brought up by Reviewer #2's. We agree that these comments strengthen the manuscript and we thank you for your thoughtful reviews.

Response: Thank you, we found the reviews useful to improve our manuscript, and we have addressed all the comments below. We also included the several discussion points highlighted by Reviewer #2 in the manuscript. 

Answers to Comments to the Author

Reviewer #1:

1) L46-49: I am not sure how this fits with the rest of the manuscript. As far as I can tell the work details degradation of DNA in soil preserved by the same method.

Response: Thank you for your comment. We agree and have removed this sentence from the Abstract.

2) L164: Isn’t the kit the MagAttract?

Response: Thank you for your comment. The kit name is “MagAttract PowerSoil DNA Kit” . It is correct in the manuscript as written. Please see: https://www.qiagen.com/us/products/discovery-and-translational-research/dna-rna-purification/dna-purification/microbial-dna/magattract-powersoil-dna-isolation-kit/#orderinginformation

3) L165: Capitalization of fungal

Response: We have addressed this.

4) L166: Capitalization of prokaryotic.

Response: We have addressed this.

5) L168: Period needed after the reference.

Response: We have addressed this.

6) L194-198: Just to be clear, the samples were rarefied to the same depth once in phyloseq and not again in vegan.

Response: Thank you for your comment. Yes, the samples were rarefied only once. We have tried to clarify this point in the text:

“Rarefaction curves were calculated in vegan with the function “rarecurve” [1]. Since different experiments showed different minimum library sizes and were independent one another (we never try to compare metrics across experiments but alway within each experiment) the dataset was divided into 3 subsets, each representing the different experiments described above, and the data matrix for each was normalized with the “rarefy_even_depth” function in phyloseq. ”

7) L208-209: Was the Bray-Curtis used as the distance?

Response: Yes, Bray-Curtis dissimilarity distance was used for the community distance metric. We have addressed this in the manuscript: “All ordination analyses were performed with Bray-Curtis dissimilarity distances [2].”

8) L217: a prior needs to be italicized

Response: This has been done.

9) L455-457: The idea of DNA degradation is due in part to environmental variation has been mentioned several times in the manuscript. I think it would be advantageous to include some data on soil physicochemistry to help support this point.

Response: Thank you for your comment. Unfortunately, we are unable to include the soil data to the analysis at this time. This because multiple reasons: 1) the soil analysis records present in the KBS archive are incomplete and the most critical years (2000 and 2005) for our experiment are missing; 2) soil analysis are not taken regularly (every year) for the Deciduous forest (DF) sites so we will miss the whole DF habitat; 3) the archived soil amount we were given for DNA extraction was very small so that the amount we have left it is not usable for chemical analysis; 4) even if we could perform a few chemical analysis on the remaining soil amounts, with COVID19 it would be extremely difficult to get these data in a timely fashion manner.

10) L463:469: Same comment here.

Response: Please see above on how we have addressed this.

11) L473-474: Consider making this a statement rather than a question.

Response: We have rephrase the section title as suggested

12) L501: What is meant by ‘microbial activity’?

Response: Thank you for your comment. We have rephrased accordingly: “Our data show that DNA degradation and microbial activity (e.g. growth, reproduction) by xerophiles comprises…”

13) L530-533: Same comments as for L455-457.

Response: We have addressed this. Please see the answer to the previous comment.

Reviewer #2: 

1) With the fungi it’s very clear that most of the drop in richness occurs between year 0 and 1 (Fig 1 A & B). In fact, the somewhat erratic behavior (year 10, DF exp 2) looks like some of the changes in later years are primarily noise. That large initial drop in richness suggests that it's the drying not the age that is causal. I suspect that is because much of year 0 signal is from mycelium, which may be more prone to autolysis as vesicles containing nucleases are ruptured. Whatever the cause, it is a striking result that is not really mentioned. It could be emphasized more when methods for archiving are brought up in the discussion.

Response: We thank you for this comment and agree. We have incorporated this interesting point into the discussion.

2) The culturing part of the experiment is pretty limited and not really well incorporated into the rest of the study. I suggest dropping it, as it adds very little. The weedy taxa found are those that are always found. I would be very surprised if others have not already reported these from archived soils, but no such studies are referenced. Dilution to end point culturing would be needed to see other taxa.

Response: Thank you for your comment. We agree with the reviewer that the culturing part of the experiment is limited but we feel it is useful to show that the soil culturable diversity declines significantly, even if the soils are still “alive”.. We rephrased all the sentences that involve the results of the culturing experiments to give less emphasis.

3) The heat maps are color-coded backwards – red (i.e, hot) is usually the highest value not the lowest. The highest intensity of red in these is counter-intuitively the least abundant/least rich, and the sea of white looks like missing data instead of intermediate values. I suggest a single gray scale going from white (0) to black (max), or white (0) to very dark red (max). The only reason for 2 colors in heat maps (e.g., gene expression data) is that usually they are referenced to increases (red) versus decreases (blue) from some starting point or control. Since there is no control or starting point used as a reference a single color would work better to communicate the patterns of change.

Response: Thank you for your comment. We changed the color scale of the heatmap according to this suggestion. Please see answer to question #4, below.

4) However abundance and richness are represented, interpreting changes in them over time is not straight-forward, and the authors should be careful on how they do this. Since these are relative abundances, they can increase over time even if their absolute abundance has decreased; they simply have to decrease less than other taxa to show an increase in relative abundance. The increased presence of Agaricomycetes in figure 4, for example, could be caused by resistant spores that lose their intact DNA at slower rates than taxa with less resistant spores. Wallemiomycetes could have exactly the same explanation, alternatively maybe these xerophiles could grow in the dried soil, but the authors have certainly not provided evidence to support that here, in spite of their indirect claims made on line 501. I think one would need qPCR to resolve this.

Response: Thank you for your comment. We agree on these points for relative abundance and discuss this possibility. Different taxa may have different extraction efficiency (the DNA concentration of a taxon after DNA extraction equals its initial cell concentration multiplied by its DNA yield per cell) that may cause a taxon to be abundant, in a sample that contain low efficiency taxa, while may be low abundant, if in another sample it appears with taxa that have high extraction efficiency. 

To solve the problem, we rescaled our read data into 0-1. In this way, even if we loose the ability to compare abundances across taxa, we can still compare abundances of the different taxa across samples. Each sample will be in the same scale but with a different 

slope. This, below, is an example on how reads and rescaled data compare.

 We have now included a paragraph in the manuscript to explain this approach: 

“To better detect taxa that showed variation across years of storage time, we first rescaled the read number of each taxa to 0-1 and then plotted it in colored heatmaps. By putting all OTUs of varying sampling depth on the same scale removes the differences in sequencing depth caused by differing library sizes between taxa (See Weiss et al. 2017 [44]).”

5) Changes in richness of particular taxa is similarly not straightforward to interpret from these data. We can assume taxon richness is a property of the sample and that it can only go down with time (i.e., we can lose taxa, but not gain them). When it appears to go up, it means that our ability to detect the richness of those taxon has changed. This may be due to increased abundance through growth, which seems unlikely but not impossible, or increased relative abundance due to loss of previously more abundant taxa. This is because as the absolute quantity of DNA drops, more of the sequences obtained are derived from the remaining resistant pool.

Response: Thank you for your comment. We included this point into the revised discussion.

6) Real differences in community across time rather than differences from sample degradation with time offer another way that time and site are not independent. The greater directional change in DF communities, might be based on this rather than higher DNA degradational changes that the authors seem to assume. In other words, the result may reflect underlying successional changes in DF, with PS less susceptible to these because it’s maintained in a narrower successional window by harvest. To tell the difference one would need archived DNA extractions. Then one could compare year 15 archived soils to year 15 archived DNA (extracted 15 years ago). This idea could be tied in the discussion of archive methods.

Response: We agree this is quite possible, and a proper experiment to test this idea is encouraged. We have included this discussion point in the revised manuscript. 

7) This idea of biological differences underlying some of the observed differences among years is discussed in the context of Tuber, Hebeloma and Cortinarius abundance changes that coincide with logging of the poplar sites but looking at the unlogged DF patterns one can also see various EM taxa wane and wax without a logging event. This should caution the authors about cherry-picked examples, and probably says there is significant year to year variation even in undisturbed forest.

Response: Thank you for your comment. We agree that this is another valid alternative hypothesis, and present as such.It is indeed true. Some taxa are not responding to the logging event, but some do so. We can say that we wanted to give some examples but if we cannot trust relative abundances because of what said above then we cannot say anything more precise here.

8) P 7 – methods make it sound like the whole ITS was sequenced for the fungi. Is that correct? It’s almost always an ITS1 or 2 spacer because the entire region is too large. In any case the real primers used are probably not ITS1f and 4, because they have none of the adapters and barcodes needed. Please correct this.

Response: Apologies for the confusion. The primers used in this study are indeed ITS1F-ITS4. We are aware that the amplified region is large, probably about 500-700 bp, depending on the fungal lineage, but these primers offer advantages. First, they are fungal specific, second, they are easily amplified compared to other primer combinations. When sequenced in a MiSeq, the two PE reads will never merge, which is the reason why only forward reads are used to perform the community analysis. This approach is valid, and the protocol has already been published numerous times.(please see [3–5] at the bottom of this document, for example). Recent studies have found that merging fungal ITS reads can lead to i) taxa losses that do not overlap well, ii) the forward reads are notoriously better quality of reverse read with the v3 chemistry, so merging the reads while maintaining high quality will generate an enormous loss of total reads and particular taxa.

9) lines 297 and 310 “at best” is used in a peculiar context – reword for clarity

Response: Thank you for your comment. We rephrased accordingly: “The models that performed the best at describing...”

Additional References now included in the manuscript 

1. Oksanen J, Blanchet FG, Friendly M, Kindt R, Legendre P, McGlinn D, et al. vegan: Community Ecology Package, R package version 2.5-6. 2019. Available: https://CRAN.R-project.org/package=vegan

2. Bray JR, Roger Bray J, Curtis JT. An Ordination of the Upland Forest Communities of Southern Wisconsin. Ecological Monographs. 1957. pp. 325–349. doi:10.2307/1942268

3. Benucci GMN, Longley R, Zhang P, Zhao Q, Bonito G, Yu F. Microbial communities associated with the black morel cultivated in greenhouses. PeerJ. 2019;7: e7744. doi:10.7717/peerj.7744

4. Longley R, Benucci GMN, Mills G, Bonito G. Fungal and bacterial community dynamics in substrates during the cultivation of morels (Morchella rufobrunnea) indoors. FEMS Microbiol Lett. 2019;366. doi:10.1093/femsle/fnz215

5. Noel ZA, Chang H-X, Chilvers MI. Variation in soybean rhizosphere oomycete communities from Michigan fields with contrasting disease pressures. Applied Soil Ecology. 2020. p. 103435. doi:10.1016/j.apsoil.2019.103435

---

## [Editor Report · Decision Letter 1]

24 Jul 2020

Patient propagules: Do soil archives preserve the legacy of fungal and prokaryotic communities?

PONE-D-20-15299R1

Dear Dr. Benucci,

We’re pleased to inform you that your manuscript has been judged scientifically suitable for publication and will be formally accepted for publication once it meets all outstanding technical requirements.

Kind regards,

Daniel Cullen

Academic Editor

PLOS ONE

Additional Editor Comments (optional):

An interesting and useful paper. Congratulations.